# The Effect of Fractional Composition on the Graphite Matrices’ Porosity

**DOI:** 10.3390/ma17215171

**Published:** 2024-10-24

**Authors:** Mariya D. Gritskevich, Alexandra V. Gracheva, Mariya S. Filippova, Maxim S. Konstantinov, Rashit R. Aitbaev, Nikolai S. Morozov, Sergei N. Chebotarev, Viktor V. Avdeev

**Affiliations:** Department of Chemistry, Lomonosov Moscow State University, Moscow 119991, Russia; gracheva.a@inumit.ru (A.V.G.); filippovams@my.msu.ru (M.S.F.); konstantinovms@my.msu.ru (M.S.K.); aitbaev.r@inumit.ru (R.R.A.); morozov.n@inumit.ru (N.S.M.); chebotarev.s@inumit.ru (S.N.C.); avdeev@highp.chem.msu.ru (V.V.A.)

**Keywords:** synthetic graphite, phenolic resin, liquid silicon infiltration, porous graphite matrices, porosity, fractional composition

## Abstract

Synthetic graphite of complex fractional composition was mixed with phenolic resin as a binder and pore-forming component. The mixtures were pressed and subsequently heat-treated to obtain porous matrices. The structural transformations of phenolic resin by heating up to 900 °C in oxygen and inert gas media were studied and the patterns of amorphization of fixed carbon formed on the walls of the pore system during carbonization were investigated. We found regularities in the changes in matrix volume density in the function of the open porosity and the average pore diameter. It is shown that, in order to obtain graphitized carbon matrices with a density of 1 g/cm^3^ and an open porosity of at least 50%, it is necessary to introduce no more than 20% of phenolic resin into the molding powder with an equal content of 60, 100 and 250 μm graphite fractions. This allows for high intensity and completeness of bulk silicon infiltration.

## 1. Introduction

Graphite is the starting material for the manufacture of articles of widespread use in different branches of industry (in 2015–2016, the total graphite production in the world was about 2.5 million tons and according to the forecast for natural graphite, the rapid growth of demand in tons will continue), such as anode materials (for anodes, graphite has a dominating market share of 98%), graphite foil gaskets, moderator materials for molten salt reactors, various composite materials, silicon carbide end seals, etc. [1,2,3,4,5,6,7,8,9,10,11,12,13,14,15,16,17,18,19,20].

Liquid silicon infiltration of porous graphite matrices gives SiC-based materials of various shapes. The patterns of the porous structure, pore size, and volume distribution of the pores affect the degree of impregnation and the final content of free silicon after silicon infiltration, which, in turn, influences the formation of defects in the material and its physico-mechanical properties [21]. The current work will be useful for creating materials from siliconized graphite based on porous graphite matrices.

As indicated in [22,23,24,25,26,27,28,29,30,31,32,33,34], the characteristics of the graphite matrix, namely, the fractional composition of the starting materials, the diameter of pores and the amount of binder affect the pore size and their distribution (areal and volumetric porosity). By varying the fractional composition and porosity of the graphite matrix, the mechanical properties of silicon carbide products can be controlled. Although there is a certain amount of work on this topic, their authors either use other source materials that require more complex and lengthy technological processes, or investigate diffusion processes occurring during the interaction of a porous graphite matrix with molten silicon, which confirms the relevance of this work. By the way, it is believed [35] that for good silicon impregnation, the density of the carbonized matrix should be on the order of ~0.963 g/cm^3^, and the volume porosity around 58%.

In this work, we studied the areal and volumetric porosity of graphite matrices with densities of 1.04–1.15 g/cm^3^, obtained from synthetic graphite powders of three fractional compositions (with particle sizes of d = 63–100 μm, d = 100–250 μm, and d = 250–315 μm) and phenolic resin (with an average particle size of d = 30 μm) in various ratios.

## 2. Materials and Methods

### 2.1. Initial Materials

GII-A synthetic graphite (Figure 1a) and SFP-012A3 phenolic resin (Figure 1b) were used as starting materials. Their characteristics are shown in Table 1.

GII-A synthetic graphite is used in metallurgy for carbonizing cast iron and steel, as well as in the manufacturing of graphitized carbon materials and products, and as a filler for graphite-reinforced plastics. It is made from waste from the electrode and metallurgical industries (scraps and stubs of electrodes, blast furnace blocks, hearth and side blocks, shaped products) in accordance with the requirements of technical specifications for fractional and chemical composition.

Phenolic resin is a kind of powdered, hardened phenol-formaldehyde resin (so-called PFR) of the novolac type (C_6_H_5_OH) and is used as a binder in the synthesis of graphite matrices for the production of silicon carbide preforms. The hardener is hexamethylenetetramine (C_6_H_12_N_4_), also known as urotropin. The hexamethylenetetramine content dictates the amount of fixed carbon [36,37]. Phenolic resin is widely used in the production of friction and abrasive materials, refractory materials, proppants, electrical insulation materials, in foundry industry, etc.

### 2.2. Sample Preparations

In this study, the following technological scheme of the preparation of graphite matrices was realized (Figure 2): the starting graphite powder was sieved and classified according to certain fractions: 63–100 μm, 100–250 μm, and 250–315 μm.

To determine the effect of the fractional composition on the porosity of graphite matrices after carbonization, several types of samples were produced which were different in the quantitative composition of fractions. Because sieving, the initial stage of fractionating graphite powder according to particle sizes, gives only a range of values, we used the laser diffraction method using the Bettersizer S3 Plus (Bettersize Instruments Ltd., Dandong, China) particle size analyzer to determine the exact fractional composition of powders. The data are shown in Figure 3.

After sieving, the resulting graphite mixture was further mixed for 1 h with phenolic resin powder (20% by mass) in PM100 Retsch planetary mill with zirconium oxide balls (d = 20 mm) providing uniform mixing without grinding the powder mixture particles.

Samples of 25 and 30 mm in diameter were pressed in a hydraulic press at 30 MPa. This pressure was selected based on the analysis of the literature [23,38]. The obtained samples were subsequently subjected to two-stage heat treatment in order to remove residual stresses from pressing, as well as for crosslinking and partial removal of the binder. For thermal treatment, the samples were heated to 175 °C in 30 min, then cured under isothermal conditions for 3.5 h. The thermal treatment stage was followed by a carbonization stage: the samples were heated to 900 °C at a rate of 2 °C/min; then, they were kept at this temperature for 30 min. All changes in geometric and physical parameters were recorded at each stage.

Thus, we obtained and examined five main samples (their starting parameters are shown in Table 2):Sample 1 (63–100 μm, 50%; 100–250 μm, 25%; 250–315 μm, 25%; +binder, 20%);Sample 2 (63–100 μm, 25%; 100–250 μm, 50%; 250–315 μm, 25%; +binder, 20%);Sample 3 (63–100 μm, 25%; 100–250 μm, 25%; 250–315 μm, 50%; +binder, 20%);Sample 4 (63–100 μm, 33%; 100–250 μm, 33%; 250–315 μm, 33%; +binder, 20%);Sample 5 (>315 μm, 100%; +binder, 40%).

During the molding process, we noted that samples with the highest content of fine fraction (50% of the 63–100 μm fraction) were the most tenacious, their edges not crumbling during the measurements.

### 2.3. Test Methods

#### 2.3.1. Areal Porosity Studies

Using an OLYMPUS BX51 optical microscope (Tokyo, Japan), the distribution of pores on the sample surface was estimated, and the areal porosity was calculated. Photomicrographs were taken at magnifications ×5, ×10, ×20, ×50, and ×100. To gather statistics, three photographs of different spots on the surface of each sample were taken—on the upper, on the middle, and on the lower part of the sample, at each magnification. Then, the most “illustrative” magnification was selected (where the pore edges were better visible). The pores were circled at each spot. Then all the pore areas (in pixels) were summed up and divided by the total area of the respective photograph, whereby the average porosity value was derived. The Altami studio v.4 program was used for image processing.

#### 2.3.2. Shape and Size of Pores in Graphite Matrices

Electronic images obtained by scanning electron microscopy (SEM) using the TESCAN VEGA 3 instrument (Brno, Czech Republic) made it possible to determine the size and shape of the particles of the source material and to examine the surface of carbonized matrices. The imaging was carried out at an accelerating voltage of 15.00 kV and WD = 10.00 mm using an SE detector.

#### 2.3.3. Structural Characteristics

Raman spectroscopy provides information on the structural characteristics of graphite matrices. To obtain the spectra, Renishaw inVia confocal spectrometer (Renishaw plc, Wotton-under-Edge, UK) equipped with a microscope, a precision XYZ table, and a 532 nm wavelength excitation laser was used. Three spectra were obtained for each sample. The laser power was 5% (0.8 MW, a spot with a diameter of 1 µm for a 50 lens). The lens for image acquisition had a field of ×50–120 × 120 µm. The shooting time was 60 × 5 (s × times). The Fityk v.1.3.1 program was used to process the spectra.

#### 2.3.4. Particle-Size Distribution

In this work, laser particle size analyzer equipped with an integrated CCD camera was used to determine the equivalent sphere diameter (D90) of GII-A synthetic graphite powder particles and their shape. The tests were carried out at 20 °C and a relative humidity of 40% with the Bettersizer S3 Plus particle size and shape analyzer (Bettersize Instruments Ltd., Dandong, China), equipped with a 532 nm wavelength semiconductor laser and a 120 frames-per-second CCD camera for dynamic image analysis. A portion of test sample (0.10–10.0 g, depending on the material density and the expected particle size) was water suspended in the container of the instrument. Surfactants were added to the water after making sure that there were no bubbles in the system, and then the analysis was performed. The test was carried out in a series of measurements, and the respective average value was taken as the result.

#### 2.3.5. Volumetric Porosity of Graphite Matrices

Volumetric porosity and gas permeability, which make it possible to draw conclusions on the distribution and size of pores in the sample, were measured by PIK-PP automated permeameter-porosimeter.

Measurements were made in a 30 mm diameter sample holder at a crimping pressure of 3.4 MPa (which was the minimum crimping pressure for this setup) and a pore pressure of 1.4 MPa.

Porosity was measured according to the Boyle–Mariotte law in the following order: gas was supplied through a pressure regulator and valves to a system consisting of a sample and technological containers (tubes, vessels, volume regulator), the volume of which is known in advance. Pressure in the system was brought to the required level (1.4 MPa), and was measured by a sensor after stabilization. Then the volume controller increased the volume of the system by a certain amount, which was also measured by the sensor. The determination of gas permeability was performed under non-stationary filtration conditions using the pressure fall-off method.

#### 2.3.6. Soot Formation Rate

Thermogravimetric analysis (TGA) consists of measuring the dependence of the mass of a solid sample on the temperature of the medium in which it is placed. In this work, TGA was carried out alternately in two different environments—oxygen and nitrogen—in an STA 449 synchronous thermal analyzer (manufactured by NETZSCH-Gerätebau GmbH, München, Germany), designed for measurements of specific heat and temperatures of phase transitions, as well as of changes in the mass of solid and powdered materials during their heating.

As a result of thermogravimetric analysis, a mass loss curve vs. sample temperature is recorded. This curve is called simple or integral and shows the entire mass loss from the beginning to the end of heating process.

## 3. Results and Discussion

### 3.1. Physico-Chemical Characteristics

To make a matrix, it is imperative to make a graphite frame. However, graphite cannot be pressed separately with different pressure forces, so various types of binder have to be added. For example, where phenolic resins are used as a dry binder, only a moderate amount of effort is required to manufacture a preform of sustainable shape (Figure 4a). Pressed phenolic resin samples were able to retain shape at as little as 5 MPa minimum pressure (Figure 4b). To binder-free graphite samples, a pressing force was applied ranging from 5 to 60 MPa in 5 MPa increments (the geometric dimensions of the samples are indicated in Appendix A). After pressing, all the resulting samples were unsustainable and developed cracks, and none of them retained their shape.

In this study, we used phenolic resin SFP-012A3 to manufacture a porous graphite matrix. Phenolic resins are thermosetting polymers in which urotropin delivers high strength. A reaction of phenolic hydroxyl group dehydration is triggered at 300 °C with the release of oxygen, followed by carbonization at about 400 °C. Consequently, pyrolysis results in a decrease in mass. At 500 °C and higher, the resin carbonizes almost entirely, retaining some hydrogen and oxygen [36,37]. As soon as carbonization is over, phenolic resin transitions into a solid state (Figure 5a).

To explore the processes concerned in phenolic resin carbonization and afterward, we used thermogravimetric analysis (TGA) to find out that its almost complete combustion in oxygen takes place at around 650 °C with an undegradable residue of less than 1% (0.05 mg, which is indicated on the TGA curve in Figure 5b). When the temperature increases up to 800–1000 °C, the polymer is destroyed, including the dehydrogenation of aromatic rings [39,40].

Other temperature ranges are recorded when phenolic resin is heated in nitrogen, which acts as a protective medium (Figure 6a,c, full video on demand). An endothermic reaction releases multiple substances at different temperatures of pyrolysis (Figure 6b) [41]. Images of the phenolic resin in the carbonation process were taken using an optical microscope with a heating table attachment capable of heating the sample in a crucible to temperatures of 1500 °C. The table was equipped with a coolant supply and discharge system, as well as an inert gas supply system. A sample of 1–2 g of phenolic resin was placed in a small crucible of the table and a standard carbonization step was performed, without exposure.

During a four-hour-long thermogravimetric testing of the phenolic resin in nitrogen, the curve never reached the plateau. It follows that complete combustion in an inert medium would take much longer. In our case, carbonization in graphite filling lasted about 12 h (7.5 h heating, 0.5 h holding and 8 h cooling).

We further checked our findings by heating phenolic resin at a rate of 15 °C/min. Monitoring the process through an optical microscope, we were able to record changes during the heating and to ascertain that the binder does not burn entirely in the absence of oxygen.

At the early stages of pyrolysis, a discharge of gases created numerous pores in the carbonized sample because of polymer structure condensation. When the pyrolysis temperature exceeded 500 °C, the polymer structures of the resin gradually turned into a glassy carbon structure and the gases could easily move through pre-existing pores without further expansion in volume [42].

By burning the phenolic resin in graphite filling, we could record changes in mass after carbonization (Table 3). PR stands for phenolic resin.

To explore how graphite interacts with phenolic resin after thermal treatment and carbonization, we made matrices with a large-fraction graphite (>315 μm) and an excess of phenolic resin (40%, 50%, and 60%) (Figure 7).

We could see that thermal treatment changed the masses and geometric dimensions in all samples; and only one sample, with 40% binder, was found to be similar, after thermal treatment, to samples with the average binder content; and it was subsequently carbonized (Appendix A). For statistical purposes, three samples were produced in each series, for which data are shown in Table 4.

As can be seen from the photos and measurement readings, an excess of phenolic resin in the composition made the sample grow notably higher and wider in diameter, whereas its mass did not change as significantly. Apparently, when the content of finer-fraction binder went up, the overall packing density inside the preform grew. This made the reaction during carbonization more exothermic and literally made the samples “burst out” from the inside. Our tests show that 40% binder content in a graphite preform may be described as the threshold in terms of whether or not the preform will be able to retain shape. On the other hand, this threshold is acceptable for studying changes in the porosity of the sample for scaling purposes.

Using SEM (for the images, see Appendix A), we could explore how the fixed carbon of phenolic resin interacted after carbonization with the graphite in the matrix. Pores were uniformly distributed on the surfaces of all the samples, and they differed in size and shape only, depending on the ratio of fractions. Where fine fractions of graphite are prevalent in the composition, angular pores of relatively small size were visible in the matrix. As the fraction grew larger, the pores became fewer but had a larger average diameter of between 50.36 and 59.64 μm. Furthermore, the geometry of the pores was different, as irregularly shaped, rounded and more oblong pores started to emerge. At a closer view, some pores were found to be hollow (with graphite lamellae at the edges) and some not, with rounded prominences at the edges (Appendix A).

### 3.2. Structural Characteristics and Composition

Raman spectroscopy is one of the key methods for exploring changes in the structure of phenolic resin and graphite-based materials [36,43]. In this study, we collected Raman spectral data for the initial materials (phenolic resin and graphite in powder) and the graphite preform with the similar graphite fraction composition (Figure 1a,b). The Raman spectrum for phenolic resin powder showed a set of narrow and wide peaks in the range from 0 to 2500 cm^−1^, corresponding to the polymer structure of phenolic resin [40].

We collected Raman spectra from several points in the sample (Figure 8). In Raman spectroscopy, the shape and position of the D and the G peak are recognized as the key characteristics of carbon material structure. The G peak (graphite) is predicated on a stretching of bonds in all sp^2^ atom pairs, in both rings and chains. The D peak (disorder) is predicated on the breathing modes of sp^2^ atoms in the rings, indicating structural defects, which it needs for its activation. Additionally, each peak has its specific “stationary” position on the spectrum: the D at ~1360 cm^−1^, and the G at ~1580 cm^−1^ [43,44,45,46,47,48,49,50,51,52,53].

In one case, the sample had graphite areas (the black spectrum) with the G peak at 1578 cm^−1^. In another, an amorphous structure could be seen emerging after phenolic resin carbonization, with two D and G peaks, quite characteristic, but with a shift from the standard position (1360 and 1595 cm^−1^, respectively). A G-peak shift along the x-axis typically indicates disorder, but in coal spectra, a peak position close to 1600 cm^−1^ was found as well [40]. This suggests a major reformation of the structure and the bonds in phenolic resin, and is recorded in the Raman spectrum. The intensity (calculated by each peak height) of the G peak is greater than that of the D band—I_D_/I_G_ = 0.91, which indicates the high density of defects. The higher this ratio, the greater the number of defects present [53,54,55].

### 3.3. Surface Morphology

Optical microscopy is a method of study of graphite matrix surfaces for assessing pore distribution on the surface and what their sizes are. In this study, we estimated the areal porosity of graphite matrices of different fractional compositions (S4–S8). For a number of reasons, however, this method is not very accurate for polished surfaces: it is impossible to distinguish a pore from a cavity formed during sample preparation, and the operator may be subjective in his conclusions. Operator error during measurement can be eliminated by making thinner cuts (Figure 9). This study included two types of cuts, thin longitudinal (L-direction) and transverse (T-direction) lamellae cuts made to assess the sizes and shapes of pores.

Pore images were different in different parts of the sample. The optical image of the surface showed long and wide pores of a complex shape, with 23.05% areal porosity (S). Pores in the T-direction, on the other hand, were elongated, narrow and much smaller in area. The reason probably lies in graphite matrix pressing during sample preparation. T-direction areal porosity was 23.8%.

Using the geometric dimensions of the samples and their masses (by the first method, which has measurement errors, but is less demanding for sample preparation), we calculated the density of graphite matrices after carbonization by the first method (Table 5):

Fixed carbon was calculated in the following way:(1)$$m_{pr}=m_{gi}\cdot \omega_{pr}$$
(2)$$∆m=m_{c}-m_{gi}$$
(3)$$\eta =((m_{pr}-∆m)/m_{pr})\times 100\%$$
where $m_{pr}$ is phenolic resin mass

$m_{gi}$ is an initial graphite matrix mass$\omega_{pr}$ is mass content of phenolic resin in the initial matrix∆m is change in mass after carbonization$m_{c}$ is mass of graphite matrix after carbonizationη is fixed carbon

Sample 1, with the prevalence of finest fraction graphite, had the lowest areal porosity, with fewer pores per unit of area than in other samples.

### 3.4. Porosity and Gas Saturation

Areal porosity describes only the size and shape of the pores and the area they occupy on the surface. To understand how porous a sample is, it is imperative to use the gas saturation method that helps assess its volumetric porosity. (It should be borne in mind, however, that only open pores can be assessed). The table below (Table 6) presents our volumetric porosity and gas permeability findings.

Built-in software used the following formula to calculate open porosity:(4)$$K_{por}(He)=\frac{V_{por}}{V_{sample}}=\frac{V_{T}\cdot (P_{2}-P_{1})+P_{2}\cdot \Delta V}{(P_{1}-P_{2})\cdot V_{por}},$$
where $K_{por}(He)$ is effective porosity, unit fraction

$V_{por}$ is sample pore volume, m^3^$V_{sample}$ is sample volume (based on its length and diameter), m^3^$V_{T}$ is the volume of technological containers, m^3^$P_{2}$ is internal system pressure after volume expansion, Pa$P_{1}$ is internal system pressure before volume expansion, Pa∆V is system volume expansion, m^3^$K_{perm}$ (*He*) is gas permeability by helium$K_{por}$ (*He*) is effective porosity by heliumb is gas channelling ratio, Pa$\delta_{d}$ (*He*) is density, g/cm^3^

The highest volumetric porosity was recorded in Samples 1 and 4, i.e., when fractions in the composition are uniform and fine graphite fraction prevails (Figure 10). Densities by helium were practically equivalent to those obtained through measuring geometric dimensions and masses.

Effective porosity increased with the density and the content of finer fractions in the graphite matrix. Table 7 shows areal, volumetric porosity and the average pore size obtained by optical microscopy, gas saturation and SEM.

The table shows that areal and volumetric porosity values correlate poorly, which suggests that the distribution of pores in the sample cannot be deduced from their position on the surface.

We developed a chart to show how the average pore size (measured with SEM) depends on volumetric porosity (Figure 11).

## 4. Conclusions

We explored the structural and physical properties of porous graphite matrices of different fractional compositions in various ratios with densities between 1.0 and 1.2 g/cm^3^ obtained by mixing, cold pressing and several stages of heating. Volumetric porosity goes down from 49 to 45% and areal porosity up from 12 to 19%, as density increases.

We have demonstrated that there is no direct correlation between areal and volumetric porosity in graphite matrices because the pores are asymmetrical and do not retain their diameter (shape) as they move deeper into the sample.

Raman spectroscopy has shown that the composition of graphite preforms after carbonization is heterogeneous, being a mixture of sections of graphite and amorphous carbon and they have some structural disorder; this is confirmed by the shape and position of the D peak and G peak, as well as the I_D_/I_G_ ratio.

Using SEM, we have been able to demonstrate that pores formed by phenolic resin are morphologically different from those which form “naturally” and consist of graphite particles which are likely to affect the thickness of the formed silicon carbide when graphite billets are spilled with silicon.

In order to obtain graphite matrices with ρ ≈ 1 g/cm^3^ density and ≈50% porosity (what is needed for a good liquid silicon infiltration), it is necessary to use a mixture of graphite with 33% of each fraction (d = 63–315 μm) and 20% of phenolic resin.

To achieve a volumetric porosity close to the value of 58%, it is probably necessary to optimize the ratio of the fractional composition (taking into account phenolic resin, since it has the smallest particle size), since the particles are stacked in a certain way during the molding process, small particles fill the space between the large ones.

In the future, it is planned to carry out liquid silicon infiltration of graphite matrices with different volumetric porosities and check how well the process of silicon shedding and the formation of silicon carbide goes.

## Figures and Tables

**Figure 1 materials-17-05171-f001:**
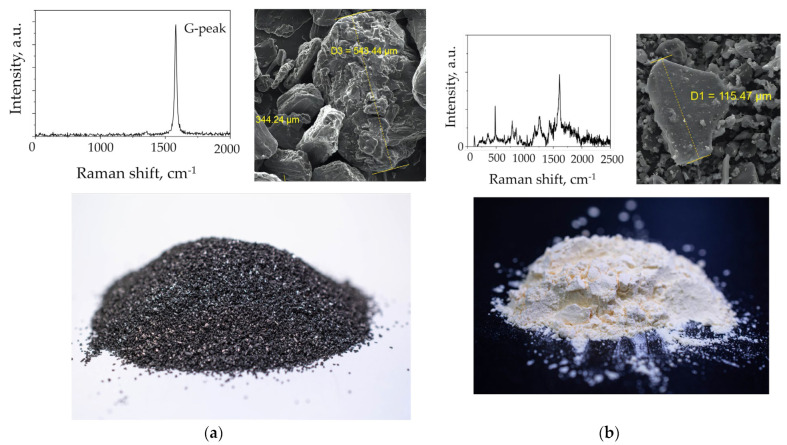
(**a**) GII-A synthetic graphite powder, its average particle size and characteristic Raman spectrum; (**b**) SFP-012A3 phenolic resin, its average particle size and characteristic Raman spectrum.

**Figure 2 materials-17-05171-f002:**
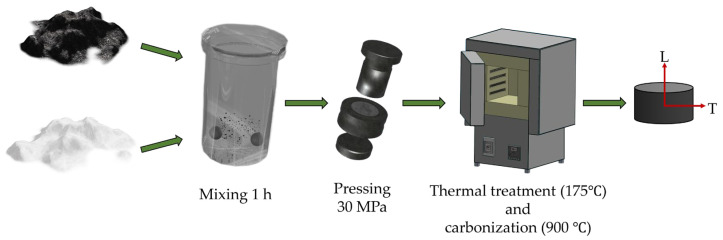
General scheme of graphite matrices preparation.

**Figure 3 materials-17-05171-f003:**
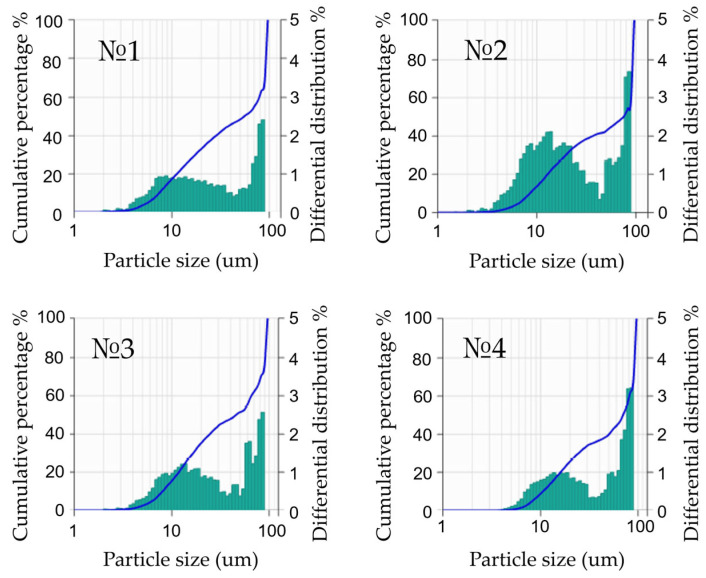
Particle-size distribution of the obtained samples, containing 20% phenolic resin.

**Figure 4 materials-17-05171-f004:**
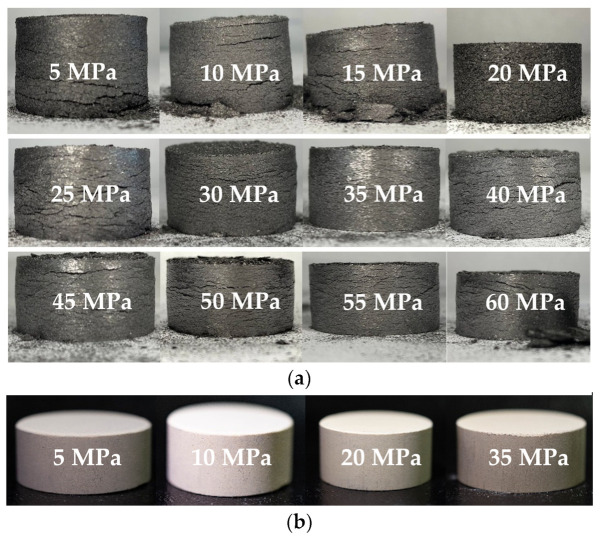
A general view of graphite matrices free from phenolic resin at different pressure forces (**a**); and compressed phenolic resin (**b**).

**Figure 5 materials-17-05171-f005:**
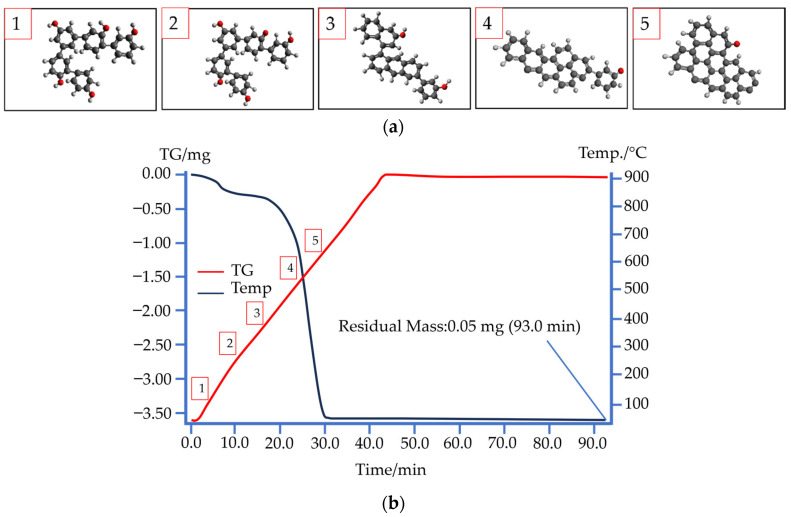
Structural changes in phenolic resin exposed to temperatures: (**a**) phenolic resin structure changing: (1) initial state, after curing; (2) dehydration begins; (3) carbonization begins; (4) carbonization continues; (5) phenolic resin carbonization near complete; (**b**) a TG curve for phenolic resin in oxygen.

**Figure 6 materials-17-05171-f006:**
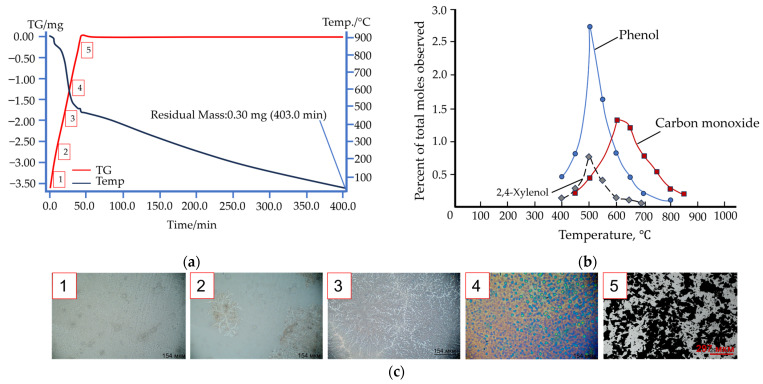
Structural changes in phenolic resin in an inert medium: (**a**) a TG curve for phenolic resin in an inert medium; (**b**) distribution of products of decomposition of phenolic resin; (**c**) phenolic resin carbonization as seen through an optical microscope: p. 1—initial state; 2—carbonization begins; 3—carbonization ends; 4—pores forming in the resin; 5—fixed carbon.

**Figure 7 materials-17-05171-f007:**
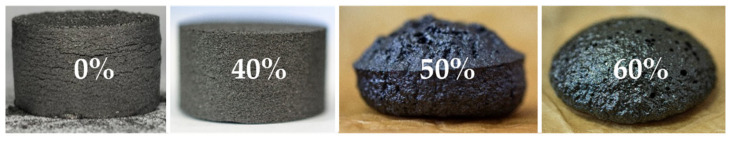
Binder content impact on the geometric parameters of graphite preforms after thermal treatment (phenolic resin as % of the total mass of the graphite matrix).

**Figure 8 materials-17-05171-f008:**
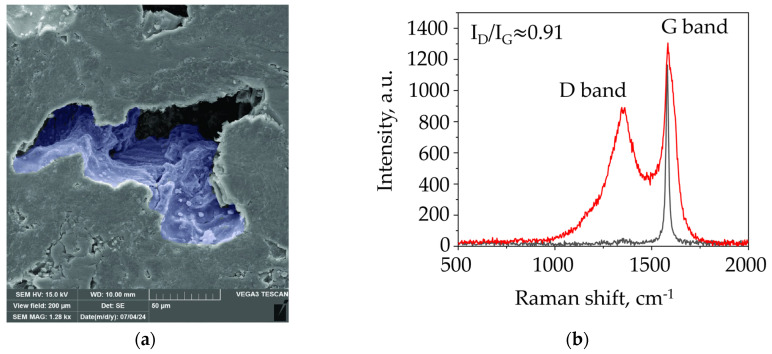
Structural features of graphite matrices after carbonization: (**a**) an SEM image of a pore with amorphous carbon inside; (**b**) Raman spectra for different sections of Sample 4.

**Figure 9 materials-17-05171-f009:**
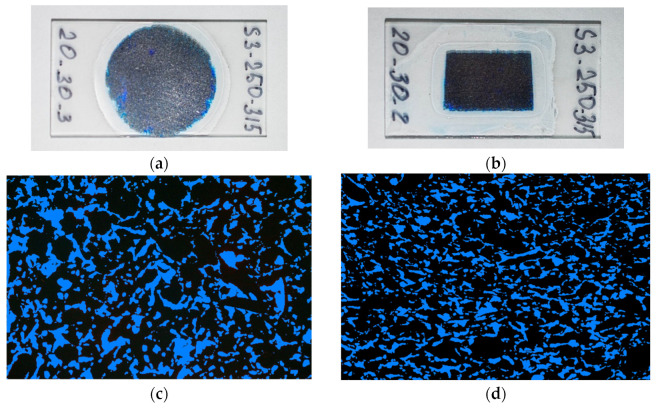
Images of sample surfaces with the pores tinted: (**a**) a general view of the surface section; (**b**) a general view of the cross section; (**c**) an image in the L-direction, S = 23.05%; (**d**) an image in the T-direction, S = 24.92%.

**Figure 10 materials-17-05171-f010:**
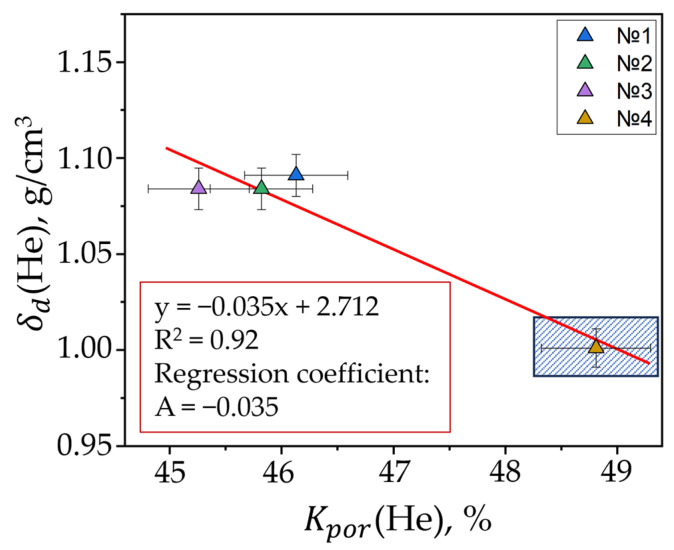
Correlation between density and porosity.

**Figure 11 materials-17-05171-f011:**
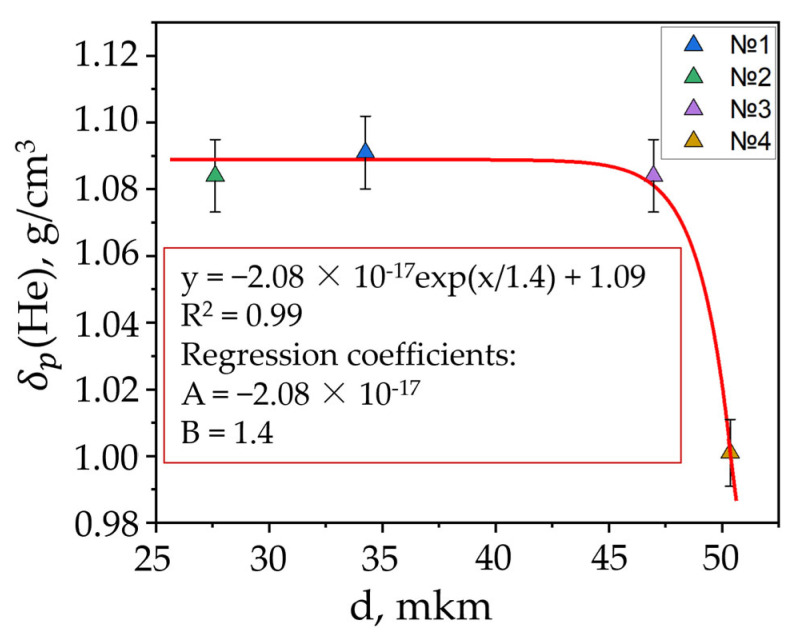
Correlation between density and average pore size.

**Table 1 materials-17-05171-t001:** Characteristics of GII-A graphite and SFP-012A3 phenolic resin.

GII-A Graphite Characteristics	SFP-012A3 Phenolic Resin Characteristics		
Parameter	Value	Parameter	Value
Ash content, Ac (%)	0.8	Fluidity, mm	45–65
Water content, W (%)	0.8	Urotropin mass content, %	6–10
Fraction, mm	0.3–1.0	Particle size, μm	0–30 μm

**Table 2 materials-17-05171-t002:** Starting parameters of graphite matrices with 20% phenolic resin content.

Sample	Mass, g	Height, mm	Diameter, mm	ρ, g/cm^3^
1 (s3_63-100_20_30_7)	9.6758	11.52	31.42	1.0838
2 (s3_100-250_20_30_7)	9.7732	11.18	31.38	1.1308
3 (s3_250-315_20_30_7)	9.9777	11.41	31.1	1.1517
4 (s3_20_30_add_2)	9.9926	12.53	31.18	1.0449

**Table 3 materials-17-05171-t003:** Tests for fixed carbon.

Sample	Pressure Force	Initial Mass	Mass after Heat Treatment	Fixed Carbon, %
PR_1	4 MT	1.67	1.03	62
PR_2	4 MT	3.00	2.03	68
PR_3	4 MT	3.05	1.83	60
PR_4	4 MT	2.17	1.30	60

**Table 4 materials-17-05171-t004:** Changes in mass and geometric dimensions of preforms after thermal treatment, depending on the content of phenolic resin.

Sample	Change in Mass, Δm, %	Change in Diameter, Δd, %
40% phenolic resin	down by ~1.70%	down by ~1.46–2.09%
50% phenolic resin	down by ~1.80–1.85%	up by ~44.33–51.61%
60% phenolic resin	down by ~1.24–2.86%	up by ~99.64–114.93%

**Table 5 materials-17-05171-t005:** Graphite matrices’ parameters after carbonization.

Sample	m, g	h, mm	d, mm	ρ, g/cm^3^	η, %	S, %
1 (s3_63-100_20_30_7)	8.69	11	30.38	1.09	49	10.93%
2 (s3_100-250_20_30_7)	8.81	11.17	30.57	1.08	51	19.77%
3 (s3_250-315_20_30_7)	8.81	11.31	30.34	1.08	42	18.19%
4 (s3_20_30_add_2)	9.01	12.33	30.64	0.99	52	12.54%

**Table 6 materials-17-05171-t006:** Volumetric porosity and gas permeability.

Sample	Porosity, %	Permeability, 10^−3^ μm^2^		Gas Channeling Ratio, MPa	Density, g/cm^3^
	*K_por_* (*He*),%	*K_perm_* (*He*), 10^−3^ μm^2^	*K_perm_* (*He* a), 10^−3^ μm^2^	b, 0.1 MPa	δ_d_ (*He*), g/cm³
1 (s3_63-100_20_30_7)	46.13	1744.69	1695.24	0.0274	1.091
2 (s3_100-250_20_30_7)	45.82	2895.70	2841.78	0.0688	1.084
3 (s3_250-315_20_30_7)	45.26	3308.26	3232.90	0.0839	1.084
4 (s3_20_30_add_2)	48.81	580.41	573.46	0.0435	1.001

**Table 7 materials-17-05171-t007:** Pore-related findings.

Sample	Areal Porosity, %	Volumetric Porosity, %	Density, g/cm^3^	Average Pore Size, μm
1 (s3_63-100_20_30_7)	10.93	46.13	1.091	34.25
2 (s3_100-250_20_30_7)	19.77	45.82	1.084	27.61
3 (s3_250-315_20_30_7)	18.19	45.26	1.084	46.97
4 (s3_20_30_add_2)	12.54	48.81	1.001	50.36

## Data Availability

The original contributions presented in the study are included in the article/Appendix A, further inquiries can be directed to the corresponding author.

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
