# Peer review of "The Effect of Fractional Composition on the Graphite Matrices’ Porosity"

_materials, 2024, doi:10.3390/ma17215171_

Round 1

Reviewer 1 Report

Comments and Suggestions for Authors

Comments on the Quality of English Language

Reviewer 2 Report

Comments and Suggestions for Authors

The authors in the present manuscript show that the synthetic graphite of complex fractional composition was mixed with phenolic resin as a binder and pore-forming component. The mixtures were pressed and subsequently heat-treated to obtain porous matrices. Structural transformations of phenolic resin under heating up to 900 ℃ in oxygen and inert gas media have been studied. Patterns of amorphization of fixed carbon formed on the walls of pore system during carbonization have been investigated. We have found regularities in changes of matrix volume density in function of the open porosity and the average pore diameter. It is shown that, in order to obtain graphitized carbon matrices with a density of 1 g/cm3 and an open porosity of at least 50%, it is necessary to introduce no more than 20% of phenolic resin into moulding powder with an equal content of 60-, 100- and 250- μm graphite fractions. The authors should address the following issues and information’s before publication acceptance in the prestigious ‘Materials’ Journal: 

1. In Introduction, authors should add a Table that compares the graphite, composite, preparation method, and characteristics with published literatures.  

2. In Introduction, authors should explain more in detail about the novelty of this study and extend the introduction?

3. In Figure 2, authors should indicate what are black and white things and the final composite?

4. In Figure 6b, how authors calculate the moles of CO2, phenol and 2, 4 Xylenol? How authors identify these components (CO2, phenol and 2, 4 Xylenol) in the samples?

5. Authors should perform the surface area analysis of some samples and study the impact graphite composition on surface area?

6. In Figure 8b, authors should add the support the characteristics of Raman D and G bands with literatures and calculate ID/IG ratio of sample 4. Authors may go through these two publications for more details and cite accordingly: https://doi.org/10.1016/j.carbon.2024.119331 & https://doi.org/10.1016/j.matchemphys.2019.122102     

Comments on the Quality of English Language

Minor editing of English language required.

Reviewer 3 Report

Comments and Suggestions for Authors

Reviewer report on the manuscript materials-3231124
Mariya D. Gritskevich et al. “The effect of fractional composition on the graphite matrices porosity”
Synthetic graphite of complex fractional composition was mixed with phenolic resin as a binder and pore-forming component. In present manuscript, the mixtures were pressed and subsequently heat-treated to obtain porous matrices. Structural transformations of phenolic resin under heating up to 900 ℃ in oxygen and inert gas media have been studied. Patterns of amorphization of fixed carbon formed on the walls of pore system during carbonization have been investigated. Regularities in changes of matrix volume density in function of the open porosity and the average pore diameter were found.
The manuscript can be accepted after minor revision.
Overall, the quality of the manuscript is good, I recommend to Authors make some corrections:

1.The assignment of the peaks in the Raman spectra (Figure 8) are not well justified. The up-to-date references (2024) should be added. I recommend Authors using the publication [Carbon 2022, 194, 52] and references there.

2.In the subsection „Test Methods”, more details should be provided, e.g. about Raman spectra collecting. resolution during X-ray absorption spectra recording should be written.

Reviewer 4 Report

Comments and Suggestions for Authors

The manuscript titled “The effect of fractional composition on the graphite matrices porosity” by Gritskevich, M.D.; et al. is a scientific work where the authors fully characterized the porosity properties of graphite combined with phenolic resi. For it, many complementary techniques were devoted in this research. This is a topic of growing importance and the manuscript is generally well-written. However, it exists some points that need to be addressed (please, see them below detailed point-by-point) to improve the scientific quality of the submitted manuscript paper before this article will be consider for its publication in Materials.

1) The authors should consider to add the term “fractional composition” in the keyword list.

2) “Graphite is the starting material for the manufacturing of articles of widespread use (…) etc” (lines 27-30). Could the authors provide quantitative data insights according to the worldwide consumption and the economic impact in the use of graphite for the above described Industrial sectors? This will aid the potential readers to better understand the significance of this devoted research.

3) Table 1 (line 65). “Value” column. The commas should be exchanged by points. The authors should fix this issue. Same comment for the Table 2 (line 108).

4) “2.3.1. Areal porosity studies” (lines 113-122). What were the software tools used by the authors to process the raw data gathered by the optical microscope? Same comment for the rest of the M&M subsections.

5) “2.3.2. Shape and size of pores in graphite matrices” (lines 123-128). Did the authors employ any contrast agent to gather the SEM images? In case affirmative, could this lead to any problem during the SEM data interpretation? Some insights should be furnished in these regards.

6) “3.1. Physico-chemical characteristics” (lines 170-259). Here, even if I agree with the findings achieved by the authors, it should be remarkable to mention how the mechanical properties at the nanoscale [1] can be altered by the graphite content of phenolic-based composites by decreasing the friction coefficients and wear rates [2].

[1] Magazzù, A.; et al. Investigation of Soft Matter Nanomechanics by Atomic Force Microscopy and Optical Tweezers: A Comprehensive Review. Nanomaterials 2023, 13, 963. https://doi.org/10.3390/nano13060963

[2] Zhang, E.; et al. Tribological Behavior of Phenolic Resin-Based Friction Composites Filled with Graphite. Materials 2021, 14, 742. https://doi.org/10.3390/ma14040742

7) Table 3 (line 227). The significant figures need to be homogenized.

8) Figure 10 (line 339). The standard deviation bars according to the tested conditions and the regression coefficient of the linear fitting should be also displayed. Same comment for the Fig. 11 (line 350).

9) “Conclusions” (lines 352-375). This section perfectly remarks the most relevant outcomes found by the authors in this work. Then, it should be also mentioned the potential future action lines to pursue the topic covered in this research.
